# CTRL-Rec: Controlling Recommender Systems With Natural Language

## Abstract

When users are dissatisfied with recommendations from a recommender system, they often lack fine-grained controls for changing them. Large language models (LLMs) offer a solution by allowing users to guide their recommendations through natural language *requests* (e.g., "I want to see respectful posts with a different perspective than mine"). However, integrating these user requests into traditional recommender systems, which focus on predicting user interaction with specific items, remains a necessary challenge to overcome for practical applications. We propose a method, **CTRL-Rec**, that allows for natural language control of traditional recommender systems in real-time with computational efficiency. Specifically, at training time, we use an LLM to simulate whether users would approve of items based on their language requests, and we train embedding models that approximate such simulated judgments. We then integrate these user-request-based predictions into the standard weighting of signals that traditional recommender systems optimize. At deployment time, we require only a single LLM embedding computation per user request, allowing for real-time control of recommendations. In experiments with the MovieLens dataset, our method consistently allows for fine-grained control across a diversity of requests. In a study with 17 Letterboxd users, we find that CTRL-Rec was positively received by users and significantly enhanced users' sense of control compared to traditional controls.

## 1 Introduction

In this work, we propose ***CTRL-Rec***, a method that integrates natural language controls into traditional recommender systems. CTRL-Rec allows for balancing between users' explicitly stated preferences and their engagement signals, and importantly, is capable of directly influencing the retrieval stage of modern recommender systems, rather than being confined to post-retrieval ranking.

**Motivation.** Natural language could serve as an intuitive and flexible interface for controlling recommender systems (Friedman et al., 2023; Malki et al., 2025). Imagine that a user could simply say "I want content that helps me learn about other perspectives," (or any of the other requests in Figure 2) and have their recommendations update immediately in real-time. These controls could act as an important counter-balance to the engagement-focused[1] nature of modern recommender systems, giving individuals greater agency to align the recommender system with reflective or aspirational preferences that aren't captured by their engagement history (Ekstrand & Willemsen, 2016; Kleinberg et al., 2024a; Morewedge et al., 2023; Lazar et al., 2024; Rezk et al., 2024; Lukoff et al., 2021; 2023; Tan et al., 2025; Zhang et al., 2022; Feng et al., 2024). For instance, a user might want to see thoughtful long-form history videos, even if haven't typically engaged with such content. More intentional curation using stated preferences could also allow users to counter divisive but undesired content amplified by engagement-optimizing algorithms (Milli et al., 2025; Rathje et al., 2024).

Large language models (LLMs) offer promising capabilities for fulfilling such requests. For example, prior work such as Malki et al. (2025) and Kolluri et al. (2026) have utilized LLMs to steer recommender systems. However, two key practical challenges remain in fully integrating natural

---

[1]Modern recommender systems are primarily based on optimizing user engagement, e.g., likes, clicks, replies, etc. Platforms do also incorporate non-engagement signals like user controls and surveys to some extent (Cunningham et al., 2024), however, current integrations are typically limited by the sparsity of existing non-engagement signals.

Figure 1: Our approach differs from standard engagement recommenders by allowing to optimize for both revealed preferences (through engagement signals) and stated preferences (through natural language requests). This allows users to explicitly control their recommendations while maintaining the benefits of engagement-based recommendation.

language with conventional recommender systems. *First*, it is essential to be able to balance user engagement with stated preferences, since users will articulate only a small subset of their preferences, requiring the system to infer others from their behavior (Malki et al., 2025). *Second*, while LLMs could be used to directly re-rank a small set of items already present in a user's feed as in Kolluri et al. (2026) and Jia et al. (2024), if the initial pool of items is too limited, then re-ranking alone may not sufficiently reflect the user's expressed preferences. Therefore, it is important to enable natural language controls that can influence the retrieval of items from the outset—a task that is computationally challenging to directly apply LLMs to, given that the retrieval stage often involves billions of items.

**CTRL-Rec.** Our work tackles both challenges. First, to balance engagement with stated preferences, we overcome the "type mismatch" between conversational and traditional recommender systems. While traditional recommenders focus on predicting user-item interactions—such as whether a user will like a specific post—natural language requests are often broad and free-form, potentially applying to many items at once (e.g., "I want to see posts that are funny and witty but not mean"). To bridge this gap, we use LLMs to simulate users' judgments of particular items based upon their natural language requests. These user-request-based predictions can then be integrated into the standard weighting of signals that traditional recommender systems optimize, allowing for a balance of stated preferences and engagement.

Second, building upon ideas from dense retrieval (Izacard & Grave, 2021; Karpukhin et al., 2020; Khattab & Zaharia, 2020), we enable real-time control by distilling LLM-generated judgments into a dual-encoder (two-tower) architecture for computational efficiency. This approach enables natural language controls to directly influence which items are retrieved, rather than being limited to the final ranking stage. While naively computing the simulated user-item judgments would require $m$ LLM queries for each new user request (where $m$ could be in the billions or trillions), CTRL-Rec instead requires only a single LLM embedding computation per user request.

In summary, our contributions are:

1. **A scalable method for integrating language-based controls** into traditional recommender systems, allowing for real-time control of recommender systems (Figure 1). Only one LLM embedding computation is required per user request.

2. **Empirical validation on MovieLens** using both genre-specific and subjective, open-ended controls. We find that our approach effectively steers recommendations according to user requests while maintaining engagement quality.

3. **Human study with Letterboxd users.** In a study with 17 Letterboxd users, we find that CTRL-Rec was positively received by users and significantly enhanced users' sense of control compared to traditional controls

## 2 RELATED WORK

**User Control of Recommender Systems.** Traditional interfaces for allowing users to better align recommender systems with their stated preferences (Milli et al., 2021; Kleinberg et al., 2024b; Agarwal et al., 2024; Yang et al., 2019) include structured controls such as "See less often" buttons or

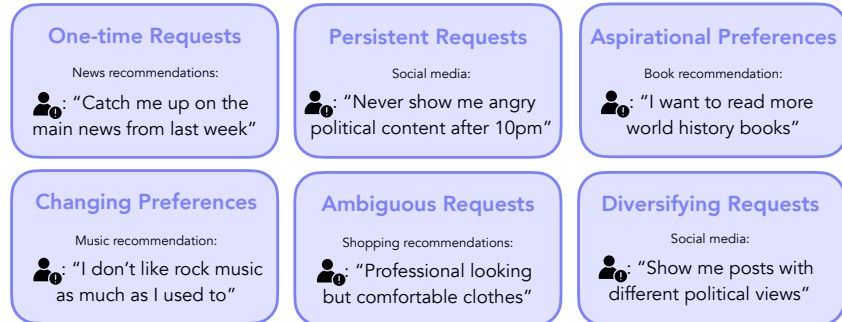

Figure 2: CTRL-Rec has the potential to be able to handle many kinds of user requests, unlocking many novel forms of user control.

genre toggles. However, research has shown that these controls are often ineffective at shaping recommendations to users' intents (Mozilla, 2023), and users frequently feel a lack of agency or control over their recommender systems (Lukoff et al., 2021; 2023). Recent work by Kolluri et al. (2026) aims to broaden user control by introducing a library of 78 value-based toggles, using LLMs to rerank social media content according to selected values (e.g., "cheerful," "knowledge"). In contrast, our approach enables users to specify their own free-form values and applies these preferences directly to the retrieval process, rather than only to post-retrieval ranking. Similarly, Malki et al. (2025) leverage LLMs to create custom Bluesky feeds. For future work, they emphasize the importance of real-time, in-context control: "Rather than treating feedbuilding as a separate setup activity, future systems should support lightweight, iterative changes directly within the feed interface." CTRL-Rec's efficiency supports this kind of immediate, interactive control.

**LLMs for Recommender Systems and Retrieval.** There has been an increasing amount of research focused on leveraging large language models for recommendation (Wu et al., 2024; Wang et al., 2024; Lin et al., 2024; Huang et al., 2024) and information retrieval more broadly (Karpukhin et al., 2020; Khattab & Zaharia, 2020). Our work sits somewhere between recommendation and retrieval, as we provide a way for users to provide natural language descriptions of their preferences which are not necessarily one-time searches (e.g. "never show me angry content about U.S. politics").

Prior research has directly used LLMs to score the relevance of items to users based on natural language preference strings (Sanner et al., 2023). However, due to the high computational cost associated with LLMs, they are limited to scoring or re-ranking only a small candidate set of items (Jia et al., 2024; Kolluri et al., 2026). More recent work in generative retrieval trains models to directly generate the identifier of the next item the user will interact with (Rajput et al., 2023; Li et al., 2025), however, this approach also suffers from high computational cost and scalability issues to large corpora (Pradeep et al., 2023). In contrast, our approach enables efficient real-time control by distilling synthetic LLM judgments. The general idea of generating synthetic labels via an LLM and then distilling has been applied before in dense retrieval (Izacard & Grave, 2021; Bonifacio et al., 2022; Huang & Chen, 2024), although with different types of judgments and for different purposes than our goal here (e.g. scoring "relevance" of answers to user questions). Note that, while in this work we distill the LLM judgments into a dual-encoder architecture (Karpukhin et al., 2020; Bromley et al., 1993), our general framework is agnostic to the exact distillation method used and could leverage other methods from dense retrieval.

The goal of our work is to give users explicit control over standard recommendation interfaces through natural language requests. In this way, our work contrasts from other research on providing recommendations limited to a chat interface (Gao et al., 2023; Friedman et al., 2023; He et al., 2023) that do not affect traditional recommender systems. It also contrasts from work that uses natural language features or profiles for recommendation (Mysore et al., 2023b; Kim et al., 2024; Paischer et al., 2024), but where these profiles are extracted or learned, rather than provided by the user as a means of control. Extracted natural language profiles are also common in more recent works on scrutable recommender systems (Radlinski et al., 2022; Penaloza et al., 2025; Ramos et al., 2024; Gao et al., 2025; Mysore et al., 2023a), which also allow for user control via editing of said profiles. However, CTRL-Rec differs both in methodology for connecting natural language to recommendations, and extends the paradigm to accommodate immediate requests (which are non-persistent), enabling more dynamic control.

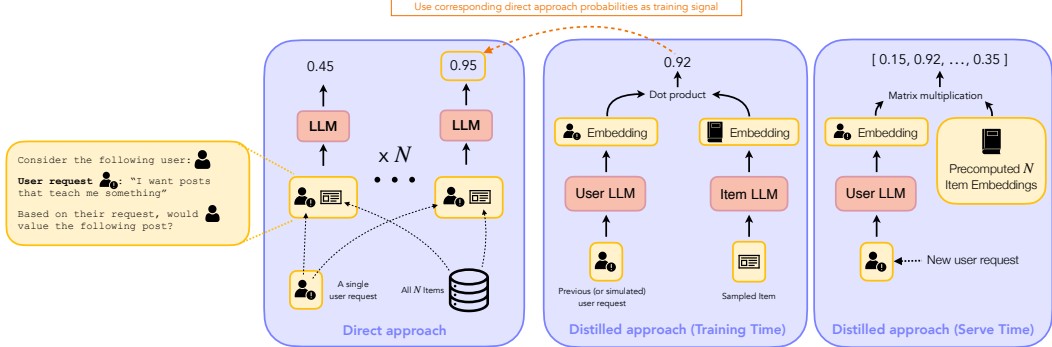

Figure 3: **Overview of our method.** We train a distilled model to approximate LLM judgements of whether a user would like a specific item based on their natural language request. This obviates the need for LLM calls (apart from a single request embedding operation) at test time.

## 3 CTRL-REC: CONTROL THROUGH LANGUAGE FOR RECOMMENDATIONS

Our system integrates natural language control into traditional recommender systems by introducing a novel preference prediction component that estimates how well items align with user-specified requests.

**Traditional Recommender Scoring.** Modern recommender systems typically optimize a weighted linear combination of different user-item signals (Milli et al., 2023; Cunningham et al., 2024; Smith, 2021; Twitter, 2023). Generally, for a given user $u$, the score of an item $i$ is computed as

$$\text{score}_{\text{base}}(u, i) = \sum_k w_k f_k(u, i) \qquad (1)$$

where each function $f_k$ returns the value of the $k$-th user-item signal and $w_k$ is the weight on that signal. The different signals $f_k(u, i)$ are typically probabilities or scores for how likely the user $u$ is to interact with item $i$ in different ways, e.g., liking, commenting, etc. Some signals may also be independent of the user $u$, for example, a signal for the likelihood that the item $i$ violates content moderation standards.

**Request-aware Recommender Scoring in CTRL-Rec.** Our key insight is that we can naturally incorporate natural language controls into this framework by explicitly incorporating the user's natural language request $r$ as an independent signal. Our updated scoring function is defined as:

$$\text{score}_{\text{CTRL-Rec}}(u, i, r) = \underbrace{\text{score}_{\text{base}}(u, i)}_{\text{revealed preference + other signals}} + w_{\text{control}} \underbrace{v(u, i, r)}_{\text{stated preference}} \qquad (2)$$

Here, the function $v(u, i, r)$ represents *how satisfied user $u$ would be with item $i$ given their request $r$*, and the parameter $w_{\text{control}}$ modulates the trade-off between engagement (the user's revealed preferences) and the user's preferences stated in natural language. The nature of the user request $r$ can be very flexible. It could be a simple immediate request or search, or represent more complex, long-term preferences that users would like the recommender system to *always* follow (e.g. "Never recommend me war movies") or only follow *under certain conditions* (e.g. "I do not want to see political content after 8pm").[2]

Platforms typically select the weights on different signals through A/B tests (Cunningham et al., 2024; Twitter, 2023; Milli et al., 2023). By tuning the $w_{\text{control}}$ weight, the platform (or potentially even the user)[3] can find an effective balance between the user's revealed and stated preferences. Also, while Equation (2) uses a linear interpolation between engagement and stated preferences to align with industry practice, other promising combination strategies are possible (Milli et al., 2021), such as threshold-based filtering or multiplicative scoring. In our experiments, directly interpolating $\text{score}_{\text{base}}$ and $v(u, i, r)$ performed worse than first ranking the movies by each metric separately and then interpolating between those ranks, so we follow this approach across our experiments.

---

[2]For the latter case, the user state $u$ must contain sufficient information.

[3]Platforms could easily expose an interface that allows the user to select how to trade-off between their language request (stated preferences) and engagement (revealed preferences).

## 3.1 Value Prediction

The key challenge for our method is estimating $v(u, i, r)$—how well an item aligns with a user's request—and doing so efficiently. A naive approach, which we call **direct LLM scoring**, is to ask an LLM directly to rate how well each candidate item matches the user's request. However, the direct LLM scoring is computationally infeasible for at-scale deployments: it requires performing $m$ LLM queries per user request, where $m$ is the number of items in the candidate pool.[4]

To address this, we can distill the LLM's judgments into a more efficient scoring function—what we call the **distilled approach**—that reduces the computation needed at deployment time from $m$ LLM queries per user request to just one LLM embedding computation. Specifically, we fine-tune LLM embedding models $f$ and $g$ to approximate LLM judgments: $v(u, i, r) \approx f(u, r)^T g(i)$. The item embeddings $g(i)$ can be computed offline, meaning that at deployment time, only one LLM embedding computation per user request (plus a matrix multiplication) is needed. Moreover, as we show in Appendix B.1 this approach can be further optimized by batching simultaneous user requests together. While the idea of using LLM embeddings for engagement prediction has been explored before, we are not aware of prior work on distilling LLMs for this specific item-value prediction task which is conditioned on user's natural language requests.

# 4 Experiments

While CTRL-Rec's advantages would be most pronounced in high-volume, rapidly changing content environments (e.g., social media, news recommendation, e-commerce), practical evaluation constraints limit our analysis. The absence of publicly available large-scale datasets that include both content and engagement signals for these domains necessitates testing in alternative settings.

In light of this, we test CTRL-Rec in the movie recommendation domain, which has standard large-scale datasets of items and user interactions. We evaluate CTRL-Rec's performance through two types of complementary experiments: (a) a reachability experiment (Dean et al., 2020; Curmei et al., 2021), and (b) a human study involving users of Letterboxd, a movie ratings platform. In the reachability experiment, we randomly select two users from the MovieLens dataset—a source user and a target user—and assess how easily the source user's recommendations can be adjusted to resemble those of the target user. This large-scale experiment evaluates CTRL-Rec's potential steering to *any* observed preferences in the dataset. By contrast, our human study is smaller in scale but focuses on whether CTRL-Rec help real users steer their recommendations with the *specific* preferences they have in mind. When referring to engagement-based recommendations, across all of our experiments we use the SAR (Sequential Association Rules) algorithm, as implemented by Microsoft's `recommender` module (Graham et al., 2019).

Figure 4 shows qualitative examples returned by CTRL-Rec for three different queries. Despite the nuanced and open-ended nature of some of these requests, CTRL-Rec effectively produces relevant results. During deployment, each query requires only one LLM embedding computation, allowing for real-time control of recommendations.

## 4.1 Training Details

CTRL-Rec was trained on 220,000 simulated user preferences generated from the MovieLens 32M dataset, which contains $\sim$32m ratings by $\sim$330k unique users across $\sim$87k movies. Our training data generation process consisted of two main stages: (1) request generation, and (2) preference scoring.

**Request Generation.** We first generated a diverse set of natural language requests by sampling users from the MovieLens dataset and instructing gemini-2.5-flash-preview-05-20 to generate requests across 10 different categories designed to span the range of ways users might express movie preferences. These categories include one-time situational requests, long-term persistent preferences, aspirational requests for personal growth, similarity-based comparisons, and various types of

---

[4]One might wonder if it is possible to simply do one LLM query per user request but ask the LLM to generate ratings for all $m$ items. When $m$ is large, this request requires a long context, particularly if the items are also represented by richer natural language descriptions as in Section 4. The long context makes this approach either infeasible or slow.

Figure 4: Qualitative examples of CTRL-Rec recommendations for three diverse natural language requests ($w_{\text{control}} = 0.995$, no recommendation history).

| Request | Request | Request |
|---|---|---|
| *"Movies with inclement weather"* | *"I'd like to see more movies with the vibe as Her but not about AI"* | *"Movies with a green ogre"* |

| CTRL-Rec Feed | CTRL-Rec Feed | CTRL-Rec Feed |
|---|---|---|
| Survive! (1976) | Garden State (2004) | Shrek 2 (2004) |
| The Day After Tomorrow (2004) | Once (2006) | Shrek (2001) |
| The Perfect Storm (2000) | Lars and the Real Girl (2007) | Kung Fu Panda (2008) |
| Deep Impact (1998) | Station Agent, The (2003) | Jumanji (1995) |
| The Ghost and the Darkness (1996) | Beginners (2010) | Hook (1991) |

filtering requests ranging from simple to complex logical operations. This approach aims at broad coverage of request types that users might realistically make, though in practice a deployed system could be trained on the actual distribution of user requests.

**Preference Scoring.** For each simulated preference, we randomly sample movies from the dataset and used Llama 3.1 70B to score how well each movie matched the specific preference request. These LLM-generated scores serve as the target preferences for the distillation step described in our method (see Appendix B for details on extracting scores from LLMs). This process was repeated across many user-preference-movie combinations to generate our final training dataset of 220,000 preference examples. See Appendix C.2 for detailed descriptions of the 10 request categories and the prompts used to generate each type.

### 4.2 REACHABILITY EXPERIMENT

In our reachability experiment (Dean et al., 2020; Curmei et al., 2021), we randomly select two users from the MovieLens dataset—a source user and a target user—and evaluate how easily the source user's recommendations can be adjusted to resemble the engagement-based recommendations of the target user using both CTRL-Rec and traditional filters (specifically, for genre and decade). We chose these particular filters to align with the existing controls available on Letterboxd (see Figure 8 for a screenshot), a popular movie ratings platform whose users we recruit for our experiments in Section 4.3.

Our experimental procedure simulates a user who has a specific target recommendation feed in mind and systematically explores different combinations of filters and LLM requests to achieve it. We compare two approaches: (1) filters only, and (2) filters combined with CTRL-Rec's natural language control.

**Feed Quality Evaluation.** For each experiment trial, we generate the top-10 recommendations for both the source and target users using our engagement-based recommender system. We evaluate recommendation quality using two metrics: (1) *cosine similarity distance* (primary), and (2) *percentage overlap* (secondary). The cosine similarity between recommendation feeds is computed as:

$$\text{sim}(F_1, F_2) = \frac{\bar{\mathbf{e}}_{F_1} \cdot \bar{\mathbf{e}}_{F_2}}{\|\bar{\mathbf{e}}_{F_1}\|_2 \|\bar{\mathbf{e}}_{F_2}\|_2} \tag{3}$$

where $\bar{\mathbf{e}}_F = \frac{1}{|F|} \sum_{i \in F} \mathbf{e}_i$ is the average embedding for feed $F$, and $\mathbf{e}_i$ represents the item embedding for item $i$. This is our primary optimization target as it measures semantic similarity between recommendation sets. However, since embedding distance is approximate and relies on the quality of the underlying embeddings, we also use percentage overlap as a secondary metric, which provides a straightforward ground-truth measure that is easy to interpret – but is much more noisy, as in many cases the percentage overlap is 0.

**Filter-Only Approach.** We evaluate various filter combinations (genre and decade) applied to the source user's recommendations. To optimistically estimate users' capacity to use filters effectively, we use a greedy approach based on the target feed's genre and decade distributions. Since movies can belong to many genres and decades, considering all possible filter combinations would be computationally expensive, and most combinations would yield empty feeds with no results. Our greedy

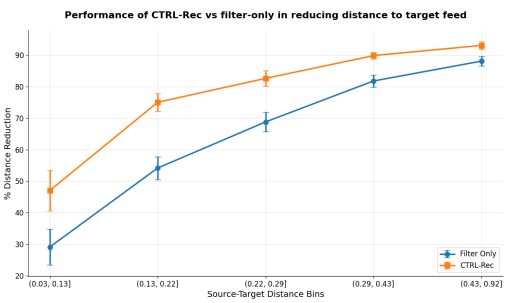 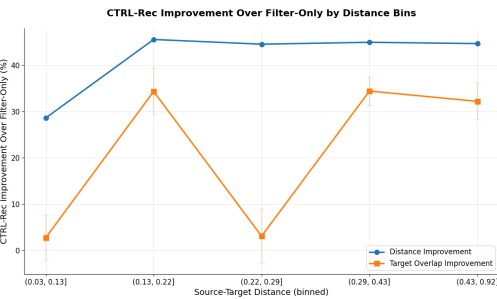

(a) Percentage of distance to target feeds achieved by different approaches.

(b) Percentage of the remaining distance to target feed cut by CTRL-Rec over the filter-only approach.

Figure 5: CTRL-Rec reachability experiment results. (a) Traditional filters can already greatly reduce distance to the target feed, but CTRL-Rec meaningfully improves performance further across all distance bins. (b) CTRL-Rec achieves up to ∼45% distance reduction for source-target pairs with higher distance, and 30% for low distances. The percentage overlap metric shows similar trends, with one outlier showing decreased overlap for high-distance pairs.

approach focuses on filter combinations most likely to yield meaningful results while still aiming to overestimate what users might realistically achieve through manual exploration. We select the combination that produces recommendations closest to the target feed in embedding space. See Appendix C.1 for details on our greedy approach.

**Filter + LLM Approach.** For the CTRL-Rec approach, we simulate an iterative user who refines their natural language requests based on feedback using Gemini Flash 2.5 as the LLM agent. We select starting filter combinations by taking the best-performing filter combination from the filter-only approach and considering all of its subsets as potential starting points. The intuition is that while the best filter combination may be optimal when filters are the only available lever, it might filter out important movies that an LLM could better target through natural language. By allowing the LLM to start from subsets (including no filters at all), we test whether natural language control can compensate for reduced filtering.

For each starting filter combination, an LLM agent: (1) observes current recommendations, (2) compares them to the target feed, (3) generates a natural language request to steer recommendations closer to the target, (4) applies the request via CTRL-Rec, and (5) repeats for up to 3 iterations. The agent refines its strategy based on previous results, simulating how users might iteratively adjust queries. For each starting filter combination, we select the natural language request (across all iterations) that maximizes cosine similarity to the target feed.

**Results.** As seen in Figure 5, we see that both filters and CTRL-Rec are very effective at bridging the gap between source and target feeds, with a large majority of the gap being closed for source-target pairs which are more distant. However, across all distances, CTRL-Rec consistently outperforms the filters. Metrics are plotted averaged across bins (with standard errors), where bins were formed by taking quintiles of the source-target distances. Moreover, Figure 5 also shows that distance reduction in embedding space (our primary measure), is accompanied by an increase in overlap with the target feed (our secondary measure), indicating that the embedding distance metric we optimized correlates with other desirable properties beyond itself.

### 4.3 HUMAN STUDY WITH LETTERBOXD USERS

Our reachability experiment provided insight into the *theoretical* controllability advantages of CTRL-Rec, but we also wanted to see whether and how these advantages surface in *real-world use*. Thus, we conducted a human study with n=17 Letterboxd users to evaluate the effectiveness of CTRL-Rec in a movie recommendation task leveraging users' own Letterboxd data. We recruited participants through social media posts, university Slack channels, and random sampling of personal connections that met the study criteria (18+ years old, have 10+ movie ratings on Letterboxd). Our study was approved by our institution's IRB, and our participants were paid 20 USD for 30 minutes.

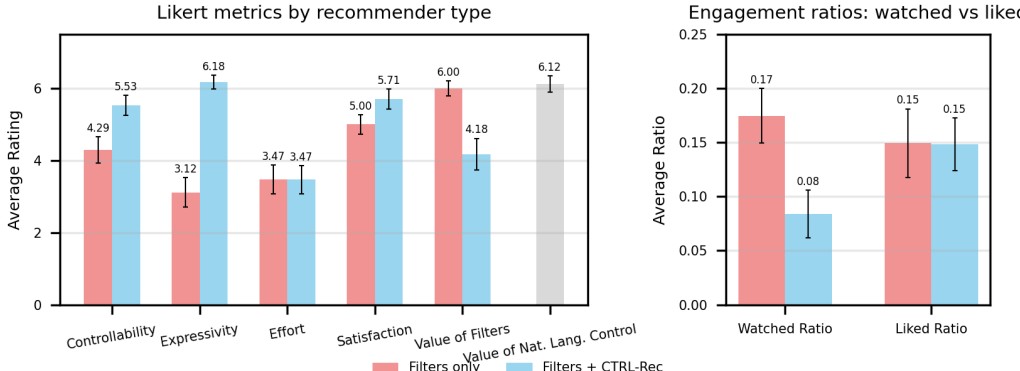

Figure 6: Human study results comparing CTRL-Rec with traditional filters to filters-only baseline. Participants found CTRL-Rec's recommendations more engaging, were more satisfied with them, and found them easier to control.

**Setup.** Our study had a within-subjects design with two cunterbalanced conditions: (a) CTRL-Rec[5] + genre and decade filters, (b) genre and decade filters only. We generated an initial set of recommendations using engagement-based recommender feed based on their ratings using the recommender in Section 4.2. Participants were then given 8 minutes per condition to use the interface to find ideal movie recommendations. For each recommended movie, we provided two buttons participants could use to indicate their engagement to the movie: "*Interested*" or "*Watched*." After each condition, participants answered Likert scale survey questions (on a 7-point scale) to rate their experience with the system. After both conditions, participants participated in a brief, semi-structured exit interview about their experiences across both conditions. Further details and study materials can be found in Appendix D.

**Quantitative results.** We compared participants' Likert scale ratings for system *controllability*[6] and *satisfaction* with recommendations, as well as *engagement*[7] using Wilcoxon signed-rank tests with Bonferroni correction. Let $W$ denote the test statistic and $M_C$ and $M_B$ denote the mean Likert rating for CTRL-Rec and the baseline, respectively. We found that CTRL-Rec was **more controllable** than our baseline (higher $M$ = more controllability) to a statistically significant degree ($p = 0.001; W = 0.0; M_C = 5.9; M_B = 3.7$). We also found that the median user was **more satisfied with their recommendation** in CTRL-Rec than the baseline (higher $M$ = more satisfied), although this was not statistically significant ($p = 0.140; W = 13.0; M_C = 5.7; M_B = 5.0$). Additionally, we found that while users were more satisfied with CTRL-Rec, their engagement – measured in terms of fraction of movies seen by the user which they clicked "Interested" for – was not meaningfully reduced. Overall, participants **agreed that the natural language controls in CTRL-Rec were valuable** ($M = 6.1 \pm 0.9$)—once they had access to the natural language controls, they reduced their reliance on filters also provided in the interface (note the drop in rating for 'Value of filters' from the baseline to CTRL-Rec in Fig. 6).

**Qualitative results.** We qualitatively analyzed participants' quotes from our studies to identify specific factors that made CTRL-Rec more controllable for users. Many participants appreciated that CTRL-Rec allowed them to **simultaneously find movies based on abstract "vibes" while also supporting highly specific queries**. For example, P5 shared that "*I understand the vibe of what I want, but I don't know that I would put it into a particular genre—that's kind of a really broad catch-all.*" P3 thought that questions like "*what's a good film about food to help explain this anthropological problem*" can't be easily answered without natural language controls. The flexibility of CTRL-Rec allowed participants to easily tune the level of specificity: P10 initially wrote a detailed query but then realized "I don't want to just be given exactly what I want" and saw value in *"deliberately underspecifying"* her preferences to the system.

---

[5]We used a $w_{\text{control}} = 0.995$ for the trade-off between engagement and stated preferences (Equation (2)).

[6]Controllability is a composite measure equally weighing survey responses to 1) how easy it was for users to control their recommendations (controllability), and 2) how much users could express their preferences through the system (expressivity).

[7]We measured engagement as the percentage of movies where they clicked "*Interested*" out of all movies recommended to them in the condition.

CTRL-Rec also allowed participants to **anchor their recommendations on other movies they've enjoyed**. P9 included *"Movies like Dune or Game of Thrones or Lord of the Rings"* in his query. Similarly, P16 described his overall strategy as: *"I was thinking of 5-star movies that I like. And I was trying to find something similar."* This resulted in recommendations that were **both high-quality and pleasantly unfamiliar**. Even P20's simple query of *"Find older movies I would like"* yielded recommendations they thought were *"pretty good, I'd never heard of [them] but seems like some things I would like."* P12 described CTRL-Rec allowed him to reach *"the outer regions of [common recommendations] where there are weird things I haven't seen but interest me,"* whereas the baseline system recommended mostly *"stuff I've heard of, or films that I'd actually watched and forgotten."* P23 echoed this, sharing that CTRL-Rec's recommendations are hard to come by with conventional search: *"I think the recommendations are really nice. I'm surprised that I don't know about these, so it's exciting. I probably wouldn't have come across them on Google."* This is also reflected in the quantitative results from Figure 6: CTRL-Rec generally showed users movies that they were less familiar with, relative to the standard engagement recommender system combined with filters (as seen by "Watched Ratio").

Finally, an interesting use case of CTRL-Rec for participants was **finding recommendations based on group preferences**. P20 queried CTRL-Rec for movies him and his partner could enjoy together, while P23, who organizes regular movie nights with friends, shared that CTRL-Rec would be valuable for finding movies that *"a bunch of friends who are also doing a PhD and who are into sci-fi can really enjoy in a group setting."*

## 5 DISCUSSION

Our results demonstrate that natural language requests can be effectively integrated into traditional recommender systems, providing users with fine-grained control while maintaining engagement quality. The success of our distilled approach suggests that LLM-based preference simulation can be made computationally efficient enough for practical deployment, requiring only a single embedding computation per user request. Findings from our human study with Letterboxd users highlight the real-world value of our method in bolstering system controllability and user satisfaction of recommended items without compromising user engagement.

**Limitations.** Our evaluation is limited to a single domain (movie recommendations), which serves as a proof-of-concept for more challenging settings where direct LLM querying would be prohibitively expensive, such as rapidly updating social media feeds or e-commerce catalogs with millions of items. Since standard LLMs can already handle movie recommendations through direct queries—albeit with significantly higher computational costs—this domain acts as a stand-in for scenarios where our efficiency gains would be more critical. Additionally, the MovieLens dataset consists primarily of well-known movies that LLMs likely encountered during training, and we do not directly test generalization to truly novel items—though this limitation is standard in retrieval systems. Our evaluation also focuses exclusively on textual descriptions, leaving questions about multimodal performance untested, though our framework could theoretically extend to models handling images, audio, or video content.

**Conclusion.** We have presented CTRL-Rec, a framework for incorporating natural language user controls into traditional recommender systems. Our key insight is that LLMs can be used to simulate how well items align with user requests, and these simulated judgments can be distilled and decomposed into efficient dot products between user-request and item embeddings. This decomposition allows us to reduce the computational cost by orders of magnitude, requiring only one LLM embedding computation per user request rather than per item. Our experiments demonstrate that this approach allows for effective steering of recommendations according to user requests while maintaining high engagement metrics. The method is computationally efficient and practical for real-world deployment. We believe this work represents a step toward recommender systems that better balance engagement optimization with explicit user preferences and control.

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

APPENDIX

**Usage of LLMs.** We used LLMs to help with rephrasing individual paragraphs of the paper for improved clarity, and writing code for the experiments.

# A    PROMPTS

## A.1    MOVIE SUMMARY GENERATION

To ensure consistent and informative movie descriptions, we pre-generate summaries for all movies using the following prompt:

---

**System:** You are a knowledgeable film critic. Provide accurate movie summaries.

**User:** You are tasked with generating a summary of a specific movie. This summary should be maximally helpful for someone deciding whether to recommend the movie to another person based on their preferences. Follow these instructions carefully:

You will be given the following information:

```
<movie_title>
{movie_title}
</movie_title>
```

Using this information, create a comprehensive summary of the movie. Your summary should focus on aspects that would be most relevant when considering whether to recommend the movie to someone. Include the following elements:

1. Basic information: Briefly mention the director, main cast, and genre

2. Plot overview: Provide a high-level summary of the plot including spoilers

3. Themes and tone: Describe the main themes explored in the movie and its overall tone (e.g., dark, lighthearted, thought-provoking)

4. Cinematic elements: Highlight notable aspects of cinematography, music, or special effects if they are particularly significant

5. Critical reception: Mention how the movie was generally received by critics and audiences

6. Potential appeal: Describe the types of viewers who might enjoy this movie (e.g., fans of certain genres, people interested in specific themes)

7. Content advisories: Mention any content that might make the movie unsuitable for certain audiences (e.g., violence, sexual content, complex themes)

Keep your summary focused and relevant. Avoid unnecessary details or trivia that wouldn't help in deciding whether to recommend the movie.

---

## A.2    GENRE USER REQUEST GENERATION

For generating user requests focused on specific genres, we use the following prompt:

---

**System:** You are an AI generating natural language commands that users might give to a movie recommendation system. You should generate a brief first-person statement about which single movie genre the user wants to see right now. Only mention one genre they want to see (not ones to avoid) and only use genres from this specific list: Drama, Comedy, Romance, Thriller, Action, Crime, Adventure, Children, Mystery, Sci-Fi, Fantasy, Horror. The genre requested should reflect what you think this user would genuinely want to watch next based on their rating history.

**User:** Consider the following movie ratings history for the user:

---

```
<movie_ratings_history>
{movie_ratings_str}
</movie_ratings_history>
```

Guidelines for the statement:

- Write in first-person perspective
- Do not mention specific movies or ratings
- Only mention one genre you want to see (not ones you want to avoid)
- Only use genres from the provided list
- The genre you request should be representative of what you would likely want to watch next, based on your rating history
- Aim for a natural, conversational tone
- Should be a single sentence mentioning exactly one genre
- Do not mention any other movie aspects besides this specific genre

### A.3    OPEN-ENDED USER REQUEST GENERATION

For generating more complex, open-ended user requests, we use the following prompt:

**System:** You are an AI generating diverse natural language commands that users might give to a movie recommendation system. You should generate a brief first-person statement about which kinds of movies the user wants to see right now.

**User:** Guidelines for the statement:

- Write in first-person perspective
- Do not mention specific movies or ratings
- Focus on preferences, likes, and dislikes related to movies
- The user may express interest in movies based on any aspect, such as genres, themes, storytelling styles, visual elements, acting, and production quality
- Aim for a natural, conversational, and informal tone, as if you were quickly expressing your preferences. Should almost always be a single sentence, potentially even a very short one
- Vary the structure and focus of the statement to add diversity

Consider the following movie ratings history for the user:

```
<movie_ratings_history>
{movie_ratings_str}
</movie_ratings_history>
```

Provide the final statement within `<statement>` tags. Here is an example output structure (do not copy the content, only the format):

```
<statement>
Final first-person statement about what kind of movie the user is currently looking for
</statement>
```

{previous_requests}

Be creative with the style of requests. Some examples of different styles:

- Direct and simple ("I want something funny")
- Descriptive ("Looking for an emotional drama that will make me think")
- Mood-based ("I'm in the mood for something thrilling and suspenseful")
- Preference-focused ("I prefer movies with complex characters and deep themes")
- Contextual ("Need a light comedy for a relaxing evening")

- Comparative ("Want something like my favorite action movies but with more humor")

MAKE SURE you open and close your statement tags correctly, `<statement>` and `</statement>`.

## A.4 DISTILLED MODEL USER REQUEST GENERATION

For generating diverse training data for the distilled model, we use the following prompt:

**System:** You are an AI generating diverse natural language commands that users might give to a movie recommendation system. The statements can include mixes of genres, or say they want to avoid certain genres, or both.

**User:** Here are some previous requests you've generated. Try to keep them diverse and focus on areas you think you may have missed.

```
<jsonl>
{requests_str}
</jsonl>
```

Please respond only with a JSONL list of 20 new request strings in a similar format, continuing the id counter from the previous requests. Do not output more than 20 new requests (stop at id 40). MAKE SURE to format the JSONL correctly. Write your response in `<jsonl></jsonl>` tags.

## A.5 FEED-LEVEL LLM JUDGE PROMPT

For evaluating how well a set of recommendations matches a user's request, we use the following prompt:

**System:** You are an expert movie recommendation system evaluator. Your task is to rate how well a list of recommended movies matches a user's request.

**User:** Consider a user with the following movie request:

```
<user_preferences>
{user_preference_text}
</user_preferences>
```

Here are the top {top_k} movie recommendations for this user:
{recommendations_str}

Rate how well these recommendations match the user's request on a scale of 1-5:

- 1 = Poor match, recommendations don't reflect user request
- 3 = Good match, many recommendations align with request
- 5 = Excellent match, all recommendations strongly align with request

Answer only with an integer from 1 to 5. Do not include any other text.

Answer:

## B EXTRACTING SCORES FROM LLM

To obtain more calibrated granular scores from LLM judgements, we follow the methodology of Williams et al. (2024). Rather than directly requesting a single rating on a scale of 1-5, we leverage the model's underlying probability distribution. Specifically, we extract the logprobs for tokens "1" through "5", normalize them into a probability distribution $P(r)$, and compute the expected rating

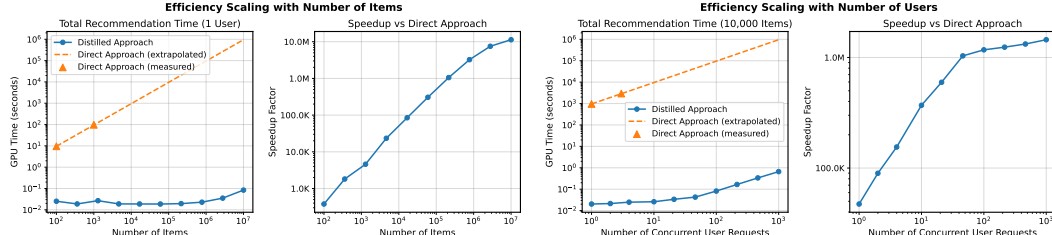

Figure 7: **Computational efficiency: the distilled approach fares far better than the direct approach at scale.** The distilled approach requires many orders of magnitude less computational resources than the direct approach for generating recommendations. Moreover, the speed-up factor for the distilled approach grows both with the number of candidate items and concurrent user requests. With $10^7$ items and $1000$ concurrent user requests, *we estimate that the distilled approach would likely be at least $10^8$ times more efficient than direct LLM scoring*. All evaluations were done on a single NVIDIA RTX A6000 GPU.

as $\sum_{r=1}^{5} r \cdot P(r)$. To verify model comprehension, we ensure the total probability mass on these five tokens exceeds 0.9.

### B.1 COMPUTATIONAL EFFICIENCY AND QUALITY OF DISTILLATION

Before considering the results of the experiments described in Section 4, we discuss the efficiency and quality of our trained distilled model.

**Distilled Model Computational Efficiency.** We benchmarked the computational efficiency of the distilled approach compared to the direct LLM scoring approach. As seen in Figure 7, the computational resources required for the direct approach grow linearly with the number of candidate items and the number of concurrent user requests. Given the high starting cost even for small item sets and a single user, it is clearly infeasible to deploy at scale without massive computational resources and parallelization. While we expect the distilled approach will also grow linearly at large scales, it's far lower starting cost makes it much more applicable to real-world recommendation scenarios.

## C REACHABILITY EXPERIMENT DETAILS

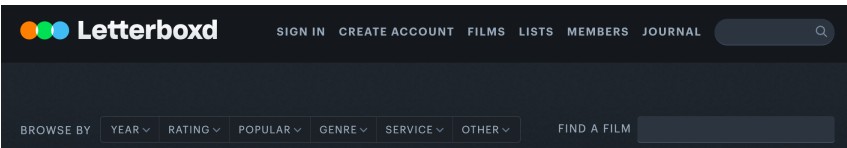

Figure 8: Screenshot of Letterboxd controls interface showing the traditional filters (genre and decade) that we compare against CTRL-Rec in our reachability experiments. We do not allow users to order by rating or popularity, as we automatically order movies using the engagement-based recommender system.

### C.1 TECHNICAL DETAILS OF FILTER SELECTION

**Genre Selection.** We analyze the target feed's genre distribution by tallying how often each genre appears across the top-10 movies, then sort genres by frequency. We construct filter combinations by incrementally adding the most common genres (e.g., Action $\rightarrow$ Action+Adventure $\rightarrow$ Action+Adventure+War), using conjunctive (AND) filters where movies must satisfy all specified genres.

**Decade Integration.** We identify the three most frequent decades in the target feed and test each genre combination both alone and with each decade individually. For example, "Action+Adventure" would be tested as: (1) Action+Adventure, (2) Action+Adventure+1990s, (3) Action+Adventure+2000s, etc. We avoid combining decades since "1990s AND 2000s" would return no results.

## C.2 TRAINING DATA CATEGORIES

Our training data spans 10 categories of user preference requests designed to capture diverse ways users express movie preferences:

1. **One-time:** Immediate, situational requests for specific moments (e.g., "need something short", "want to cry")

2. **Long-term:** Persistent preferences and standing rules (e.g., core values, content boundaries)

3. **Aspirational:** Requests for personal growth and cultural enrichment (e.g., "help me develop appreciation for...")

4. **Changing:** Evolving tastes transitioning from old to new preferences (e.g., "I used to like X, now I want Y")

5. **Ambiguous:** Poetic, impressionistic requests using metaphor and mood (e.g., "something gentle")

6. **Similarity-based:** Comparative requests referencing specific films (e.g., "like Blade Runner but comedic")

7. **Smart-filtering:** Highly specific requests requiring deep content knowledge (e.g., cinematography techniques)

8. **Smart-filtering-easy:** Clear, specific requests in simple language (e.g., plot elements, character types)

9. **Logical-filtering:** Precise logical filtering with boolean operators (e.g., "Horror from 1990s BUT NOT zombies")

10. **Refinement:** Adjusting recommendations relative to recent patterns (e.g., "more diverse than lately")

# D USER STUDY DETAILS

**Participants.** We conducted a human study with 17 Letterboxd users to evaluate the effectiveness of CTRL-Rec in a movie recommendation task leveraging users' own Letterboxd data. We recruited participants through social media posts, university Slack channels, and random sampling of personal connections that met the study criteria (18+ years old, have 10+ movie ratings on Letterboxd). Participant demographics are shown in Fig. 9. 5 participants identified as female and 12 as male. The mean participant age was $27.7 \pm 4.0$ (min: 21, max: 36). Most participants watched 3–5 movies per month on average. Our study was approved by our institution's IRB and informed consent was obtained from all participants.

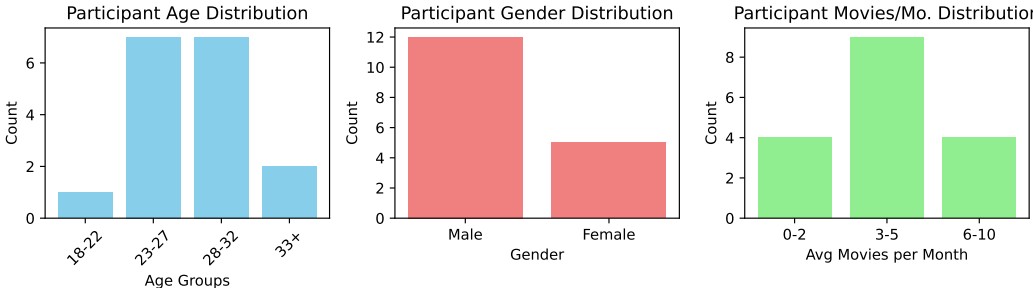

Figure 9: Distribution of age, gender, and average number of movies watched per month among Letterboxd users in our human study.

**Set-up.** Our study had a within-subjects design with two cunterbalanced conditions: (a) CTRL-Rec[8] + genre and decade filters, (b) genre and decade filters only. We generated an initial set of

---

[8]We used a $w_{\text{control}} = 0.995$ for the trade-off between engagement and stated preferences (Equation (2)).

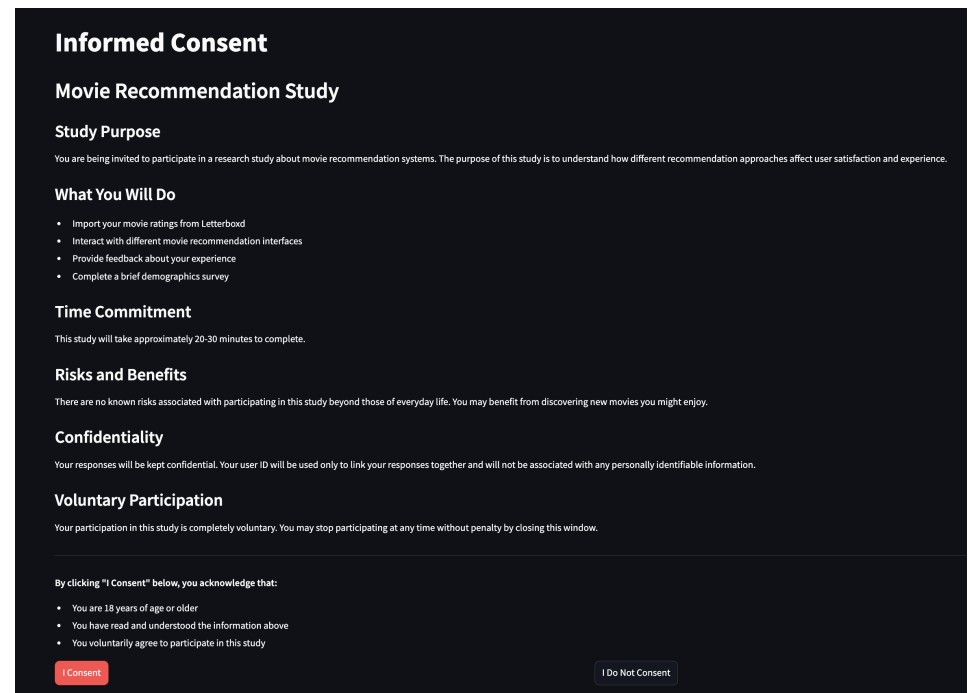

Figure 10: **User study, consent form.**

recommendations using engagement-based recommender feed based on their ratings using the recommender in Section 4.2. Participants were then given 8 minutes per condition to use the interface to find ideal movie recommendations. For each recommended movie, we provided two buttons participants could use to indicate their engagement to the movie: "*Interested*" if they were interested in watching the movie or "*Watched*" if they had already seen it. After each condition, participants answered Likert scale survey questions (on a 7-point scale) to rate their experience with the system. After both conditions, participants participated in a brief, semi-structured exit interview about their experiences across both conditions. Figures 10-12 show screenshots of our custom user study interface.

**Survey questions.** We asked users six Likert survey questions, where participants could pick from 1 (Strongly Disagree) to 7 (Strongly Agree). The following five were asked after each condition:

1. **Controllability**. "It was easy to control my recommendations"

2. **Expressivity**. "This system allowed me to articulate preferences in an expressive way."

3. **Effort**. "I had to put in a lot of effort to find recommendations I like."

4. **Satisfaction**. "I am satisfied with my recommendations."

5. **Value of filters**. "I found the genre and/or decade filter valuable."

Finally, at the end of both conditions, one with the natural language controls and one without, we asked participants how valuable they thought having the natural language controls were.

6. **Value of natural language**. "I found the natural language controls to be valuable."

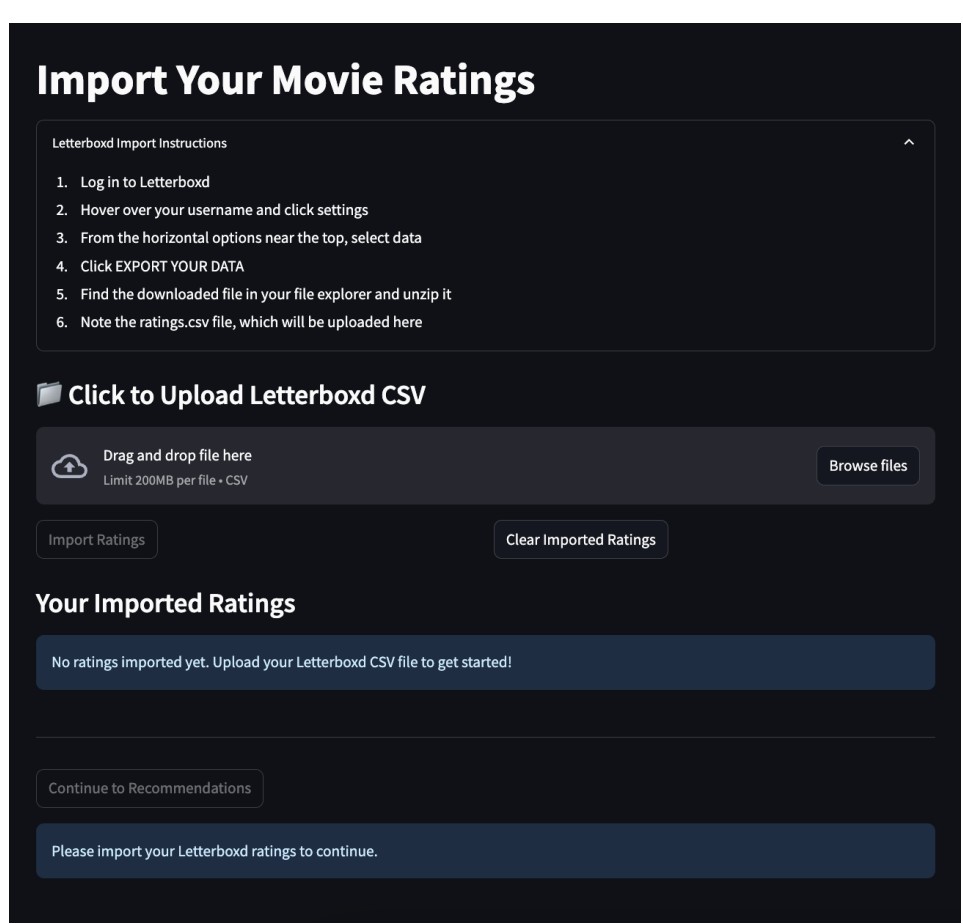

Figure 11: **User study, movie import page.**

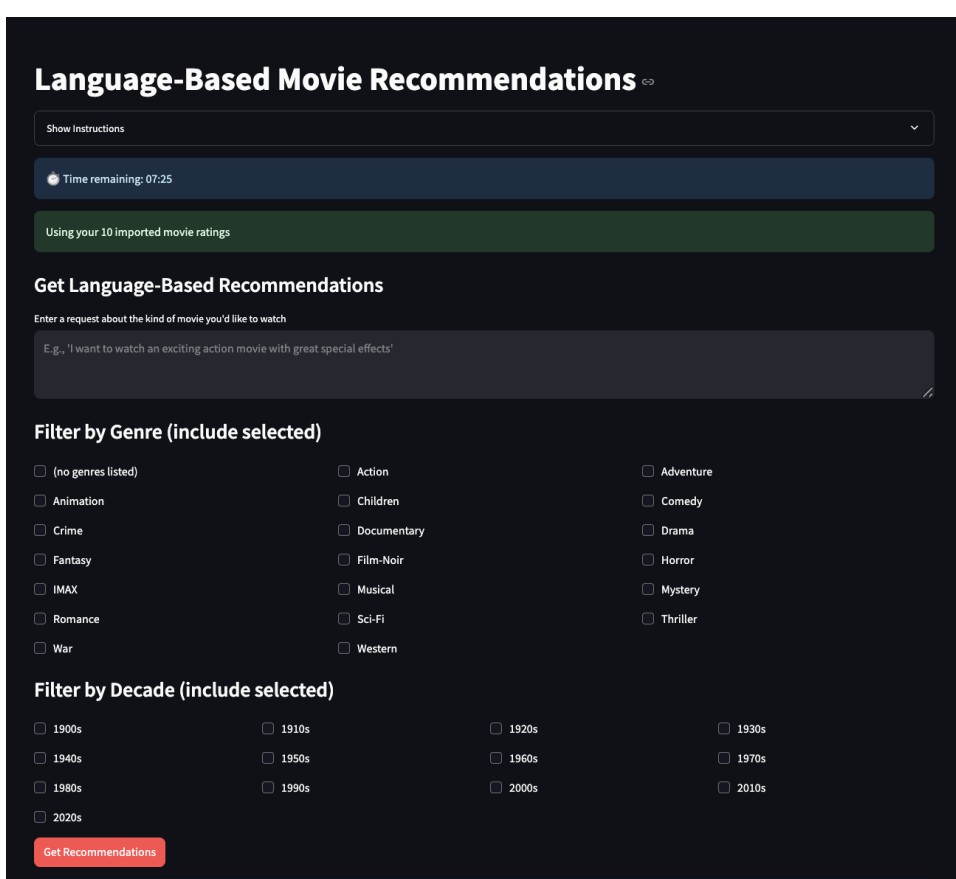

Figure 12: **User study, CTRl-Rec condition.** The condition with only the genre / decade filters is the same, except without the language text box.

