# OpenReview forum: "CTRL-Rec: Controlling Recommender Systems With Natural Language"
_ICLR.cc/2026/Conference — Submitted to ICLR 2026_

### Official Review · Reviewer_PSZP · 2025-10-27

**Soundness:** 3
**Presentation:** 3
**Contribution:** 2
**Rating:** 4
**Confidence:** 4

**Summary:**

The paper proposes CTRL-Rec, a method integrating natural language controls into traditional recommender systems to solve users’ lack of fine-grained control. It uses LLMs to simulate user-item approval based on language requests during training, distills these judgments into an efficient dual-encoder model, and requires only one LLM embedding computation per request for real-time deployment.

**Strengths:**

1. At deployment, CTRL-Rec only requires a single LLM embedding computation per user request (instead of repeated queries for massive items), enabling real-time control of recommendations.

2. Unlike traditional structured controls (e.g., genre toggles), CTRL-Rec supports free-form natural language requests (e.g., "movies like Dune but with more humor" or "gentle content for relaxation"), covering diverse needs such as abstract "vibes," group preferences, and logical filtering.

**Weaknesses:**

1. The evaluation is only conducted in the movie recommendation domain (using MovieLens and Letterboxd data), serving as a proof-of-concept but not verifying effectiveness in more complex scenarios (e.g., real-time social media feeds or large-scale e-commerce catalogs) where efficiency gains are more critical .

2. The MovieLens dataset mainly includes well-known movies that LLMs likely encountered during training. CTRL-Rec is not directly tested for generalization to truly novel items (a common limitation in retrieval systems but still a gap for practical deployment) .

**Questions:**

1. How much performance degradation does CTRL‑Rec exhibit compared to directly using an LLM for filtering?

2. Does this approach fail to handle cold-start items and rely on domain knowledge introduced by the pretrained LLM?

---

### Official Review · Reviewer_Ttkw · 2025-10-29

**Soundness:** 3
**Presentation:** 3
**Contribution:** 2
**Rating:** 4
**Confidence:** 4

**Summary:**

This paper focuses on leveraging natural language as an interface for directly controlling recommender systems. It proposes CTRL-Rec, a framework that incorporates a request-aware recommendation score—computed via the embedding similarity between user intentions and items—into conventional recommender models. To achieve computational efficiency, the authors distill LLM-generated preference judgments into embedding models. The motivation behind CTRL-Rec is that natural language can express users’ reflective intentions beyond those captured by engagement-optimizing algorithms. The authors conduct empirical experiments on the MovieLens dataset and a user study with Letterboxd participants, demonstrating that natural language requests can steer recommendation outcomes.

**Strengths:**

S1: **Scope**: This paper focuses on a crucial problem as well as a promising research direction in the recommendation field: incorporate nature language intention to interact realtime with recommend system.
S2: **Experiments**: The paper provides both empirical validation and a user study, demonstrating that LLM-based embeddings can capture human interests while maintaining engagement quality. The empirical experiment simulates how users adjust their recommendations to resemble those of a target user, comparing traditional genre/decade filters with the semantic embedding–based natural language control of CTRL-Rec. Furthermore, the authors conduct a human-in-the-loop simulation to validate the effectiveness and usability of their approach.

**Weaknesses:**

W1: Lack of Quantitative Evaluation of User Intention Embeddings.
The paper does not provide a clear quantitative analysis of how well the learned embeddings capture user intentions. Most experiments rely on heuristic or qualitative observations, making it difficult to assess whether the model truly encodes intention-related semantics beyond simple content similarity.

W2: Methodological Validation
The experimental validation does not fully support the claimed methodological contributions, as no ablation or baseline comparison is provided. CTRL-Rec comprises two core components: (1) the Request-aware Recommender Scoring mechanism and (2) the distilled embedding approach for efficiency. However, the first component’s effectiveness has already been validated in prior work [1,2,3]. Moreover, intention-based studies [3] have demonstrated that advanced language representations can already capture user preferences and refine recommendations. Therefore, it is crucial to include an ablation on distill approach or baseline comparison [1,2,3] to isolate the contribution of the distillation mechanism and to demonstrate its empirical efficiency and necessity.

W3: Lack of Experimental Transparency
The experimental section lacks sufficient detail for replication. The authors do not release implementation details, such as the specific embedding model architectures, training objectives, or hyperparameters used. The absence of released code or model specifications makes it difficult to reproduce results or verify design choices.

[1] Bi, Keping, Qingyao Ai, and W. Bruce Croft. "A transformer-based embedding model for personalized product search." Proceedings of the 43rd International ACM SIGIR Conference on Research and Development in Information Retrieval. 2020.

[2] Ai, Qingyao, et al. "Learning a hierarchical embedding model for personalized product search." Proceedings of the 40th International ACM SIGIR Conference on Research and Development in Information Retrieval. 2017.

[3] Sheng, Leheng, et al. "Language representations can be what recommenders need: Findings and potentials." arXiv preprint arXiv:2407.05441 (2024).

**Questions:**

Refer to Weakness

---

### Official Review · Reviewer_su9B · 2025-11-02

**Soundness:** 1
**Presentation:** 2
**Contribution:** 2
**Rating:** 2
**Confidence:** 3

**Summary:**

This study introduces a framework (CTRL-Rec) that integrates users' textual requests into traditional recommender systems. One of the major findings of the paper is that they find LLMs can be used to simulate how well items align with user requests, and how these simulated judgments can be distilled.

**Strengths:**

(1) Writing is easy to follow.

(2) The human study is valuable (although it would be better if the sample size can be larger).

**Weaknesses:**

(1) The differences between the proposed method and conversational recommender systems are not explained clearly. From lines 151 to 161, the paper discussed the differences, but the reasons are not compelling. "Our work contrasts from other research on providing recommendations limited to a chat interface that do not affect traditional recommender systems" From this statement, it seems that one contribution is that the proposed method affects traditional recommender systems. However, why it is necessary to affect traditional recommender systems is not clear. Why can't conversional recommender systems achieve similar effect (having controllable recommendations; experimental comparisons should be made)? Filter-only baselines are simple which may not be sufficient to demonstrate the effectiveness of the proposed method.

(2) The novelty of the method which was claimed as scalable for integrating language-based controls into traditional recommender systems could be limited. It is common these days to use LLMs to generate either embeddings or natural language descriptions of users or items, and then train traditional recommender systems with them. This line of works normally only makes one LLM call per request as well. It is easy and straightforward to add user request into the original user side augmentation too.

(3) The validity of the reachability experiment can be strengthened. Two random users are selected to serve as the "scale". Would the original difference in preference between these two users affect the experimental results? It seems expected that it is easier to adjust recommendations for the two users who have similar preferences. I understand that if all compared baselines are facing the same set of testing data, they are still comparable. However, my concern is that whether the findings would be meaningful. For example, it is possible that model behaviors changed significantly when the testing data are difficult. Another concern is that such selection is not "authentic". In other words, the "request" was engineered instead of observed.

**Questions:**

Please see details in Strengths and Weaknesses.

---

### Official Review · Reviewer_nVyf · 2025-11-04

**Soundness:** 3
**Presentation:** 3
**Contribution:** 2
**Rating:** 4
**Confidence:** 3

**Summary:**

This paper proposes CTRL-Rec, a framework that injects fine-grained, real-time natural-language control into traditional engagement-driven recommenders. By distilling LLM-simulated user-item preferences into a dual-encoder that scores requests against pre-computed item embeddings, it enables retrieval-stage steering with only one LLM forward pass per query.

**Strengths:**

1. CTRL-Rec pioneers the distillation of open-ended natural-language requests into a lightweight dual-encoder that can directly steer large-scale retrieval, bridging conversational preference and traditional engagement-based recommenders.

2. The authors test the proposed method with large-scale reachability simulations, genre and subjective open-ended requests, and a within-subjects user study, offering both quantitative metrics and qualitative user feedback.

**Weaknesses:**

1. The paper lacks an overall system-level comparison: it shows that CTRL-Rec can steer recommendations, but never benchmarks its final top-k quality against standard engagement-optimized or LLM-based recommenders under the same task. Have the authors considered adding an overall comparison and comparing it with existing recommendation baselines?

2. It is unclear why Figure 2 discusses the **potential capability** of the proposed method in the main text. If the proposed method possesses this capability, the authors need to explicitly demonstrate it through experiments.

3. The paper only validated the algorithm in the movie recommendation scenario. Have the authors considered conducting experiments in more scenarios? This would better demonstrate the algorithm's generalization ability.

**Questions:**

See Weakness.

---

### Meta-Review · Area_Chair_MH9L · 2026-01-06

**Summary:**

Reviewers recommended rejection primarily due to the lack of a comprehensive system-level evaluation, noting that the paper fails to compare its final top-k recommendation quality against standard engagement-based or LLM-based systems. A significant concern is the absence of quantitative analysis regarding user intention embeddings (W1), as the evaluation relies heavily on heuristics rather than rigorous metrics to prove the model captures intention beyond simple content similarity. Furthermore, the methodological contribution is questioned (W2); since the scoring mechanism resembles prior work , the failure to provide ablation studies or baseline comparisons prevents the isolation and validation of the proposed distillation mechanism. Consequently, the potential capabilities illustrated in Figure 2 remain experimentally unproven.

**Reviewer Concerns:**

All raised issues remain unresolved.
The authors did not submit a rebuttal. Therefore, the critical deficiencies regarding the lack of quantitative evaluation for user embeddings, the absence of necessary ablation studies to validate the distillation component, and the missing system-level comparisons against state-of-the-art baselines were not addressed.

**Reviewer Scores:**

none

---

### Decision · Program_Chairs · 2026-01-26

Reject